# Comparison Study of Diagnosis and Treatment Planning for Dental Infections between Dental Students and Practitioners

**DOI:** 10.3390/healthcare10081393

**Published:** 2022-07-26

**Authors:** Se-Lim Oh, Deborah Jones, Jong Ryul Kim, Seung Kee Choi, Man-Kyo Chung

**Affiliations:** 1Department of Advanced Oral Sciences and Therapeutics, University of Maryland School of Dentistry, Baltimore, MD 21201, USA; djones1@umaryland.edu (D.J.); cchoi@umaryland.edu (S.K.C.); 2Department of Endodontology, Temple University, Kornberg School of Dentistry, Philadelphia, PA 19140, USA; jkim@temple.edu; 3Department of Neural and Pain Sciences, University of Maryland School of Dentistry, Baltimore, MD 21201, USA; mchung@umaryland.edu

**Keywords:** periodontal abscess, periapical abscess, differential diagnosis, education

## Abstract

This study aimed to access the knowledge in diagnosing dental infections and the practice in treatment planning for the affected teeth among dental practitioners (DPs) and senior (final-year) students. A survey questionnaire containing two cases (Case A; periodontal abscess and Case B; periapical abscess) with four questions per case was delivered to potential participants. Fifty-nine DPs voluntarily participated in the survey. For senior students, the case study was a part of their course requirements; one of the two cases (either Case A or B) was randomly assigned to the 126 seniors. The distribution of responses was significantly different between the DP and senior groups except for the diagnosis of Case B (Fisher’s exact test; *p* = 0.05). Only 31% of the participants diagnosed Case A as periodontal abscess; most of them selected periodontal surgery as the first treatment option. Despite a high agreement in diagnosing Case B, the choice of treatment was significantly different; the most frequent treatment option was extraction (51%) from the DP group and root canal retreatment (57%) from the senior group. The study revealed that the diagnosis of periodontal abscess was more challenging than that of periapical abscess among dental professionals.

## 1. Introduction

Localized dental infections arise from bacterial invasion to the pulp or periodontium of the affected teeth, and are often accompanied with pain, swelling, and/or abscess [1,2]. Dental practitioners (DPs) must be able to establish a correct diagnosis for acute dental infection to provide the rapid relief of pain or discomfort for patients. A thorough understanding of both pulpal and periodontal disease processes and a correct interpretation of clinical and radiographic findings aids clinicians in making a diagnosis, determining a reasonable prognosis, and initiating the appropriate therapy.

Apical periodontitis/periapical abscess and periodontal abscess are the most common infectious conditions, considering that dental caries and periodontal disease are the most prevalent oral diseases. Untreated apical periodontitis, which is an inflammatory lesion in the periradicular tissue caused by microbial infection within the root canal of the affected tooth, can lead to periapical abscess [2]. Periapical abscess is an inflammatory reaction to pulpal infection characterized by gradual onset and an intermittent pus discharge via an occasional sinus tract; it is associated with a typical sign of osseous destruction such as periapical radiolucency (PARL) [2]. 

Periodontal abscess is an acute periodontal condition in which a pus accumulation occurs within the gingival wall of the periodontal pocket [3]. The prevalence of periodontal abscess is high among periodontitis patients who are either under active periodontal treatment (14%), in maintenance (4%), or untreated (60%) [4]. While periodontal abscess is most likely associated with pre-existing periodontal pockets in periodontitis populations, it can also develop in the absence of periodontitis from foreign body impaction, root damage due to perforation by an endodontic instrument, infection of lateral cysts, and local factors affecting root morphology such as cemental tear or crack [3]. 

To render a proper treatment, dental infection of the pulp and the periodontium must be differentiated from each other. However, making the accurate differentiation between pulpal and periodontal lesions can be challenging because the localized pulpal and periodontal conditions may exhibit similar clinical and radiographic manifestations, such as deep probing depth(s), swelling, pus discharge, tooth mobility, and osseous destruction [5]. Furthermore, determining the prognosis for teeth with dental infections can be more challenging than making the correct diagnosis of the affected teeth.

A study reported that less than 50% of senior (final-year) dental students recognized periodontal abscess during case-based learning sessions; 40% of participating senior dental students determined the prognosis for a tooth with periodontal abscess as hopeless [6]. Meanwhile, 87% of the participating students made a correct diagnosis for chronic periapical abscess with a previous root canal therapy (RCT); 60% of the students selected a root canal retreatment when the size of PARL became larger and chronic periapical abscess was developed after the initial RCT of the tooth [6]. The correct diagnosis of the disease relies on many factors, such as the knowledge, experience, and critical thinking skills of DPs [7,8]. The ability to distinguish the origin of the infection and to evaluate the severity of a disease is critical for DPs to decide whether to treat, to extract, or to refer the case to an appropriate specialist [7]. 

The purpose of this cross-sectional study was to access the knowledge in diagnosing teeth exhibiting signs and symptoms of dental infections and the practice in treatment planning for the affected teeth among DPs and senior dental students as future dental providers. This study also explored differences in clinical reasoning processes between DPs and senior dental students. The findings from this observational study will provide dental educators valuable insights to reduce errors in clinical and radiographic data interpretation among dental students and to design continuing education for DPs.

## 2. Materials and Methods

### 2.1. Ethical Approval

This cross-sectional study using a survey questionnaire was conducted under a protocol approved by the institutional review board (IRB) at the University of Maryland, Baltimore (HP-00093192). Obtaining the informed research consent was waived by the IRB because this study involved no more than minimal risk, did not affect the rights and welfare of the participants, and no personal identifiers were linked with data collection.

### 2.2. Developing a Survey Questionnaire

An expert panel, comprised of a periodontist, an endodontist, a prosthodontist, and a basic scientist, reviewed two cases that were selected by the principal investigator (PI), Se-Lim Oh. The two cases were “real-life” cases to represent periodontal abscess (Appendix A) and periapical abscess (Appendix A). Case presentations contained a brief medical and dental history, clinical and radiographic evaluation without personal identifying information. The panel formulated four questions for each case and determined the correct diagnosis. The questionnaire is available as Appendix A.

The survey questionnaire for DPs contained questions about gender and educational background information in addition to the two cases (Cases A and B) with eight questions (four questions per case; Table 1). It was delivered via a secure, web-based application (Qualtrics XM, Provo, UT, USA) to potential participants. For DPs, participation in this survey was voluntary. Once the participants opened the survey link, they had two weeks to complete it. For senior dental students, the case study was a part of their course requirements; one of the two cases (either Case A or B) was randomly assigned to the seniors. Seniors also had two weeks to complete the questions associated with the assigned case, using any available resources. 

The questions asked (1) the diagnosis; (2) the prognosis; (3) the primary determinant for assigning the prognosis; and (4) the most appropriate treatment for each case. The list of choices was provided for each question and included the option to select other in case the participants could not find their choice among the choices provided. With respect to the prognostic system, the McGuire and Nunn prognosis system [9] was provided for Case A because the system uses clinical and radiographic parameters that are commonly obtained in routine practice. The Kwok and Caton system [10] was provided for Case B because the McGuire and Nunn system does not describe any parameters associated with endodontic status. 

### 2.3. Study Participants

126 seniors at the University of Maryland School of Dentistry (UMSOD) and 59 DPs from the state of Maryland participated in this study. Table 2 summarizes the participants in this study. All didactic lectures regarding diagnoses of dental conditions are delivered by the end of the third year and dental students start patient care at the clinic in their third year at the UMSOD. There were no differences in gender distribution and the mean final grades of the third-year periodontics between the senior 1 (80 ± 12) and senior 2 (79 ± 10) groups. Case A was assigned to the senior 1 group; Case B was assigned to the senior 2 group. The DP group included 39 males and 20 females. Twenty-three DPs (39%) did not have postgraduate training, 17 DPs (29%) had advanced general dentistry or general practice residency training, and 17 DPs (29%) had training in either endodontics (9) or periodontics (8). A total of 122 participants were included for each case. A priori power analysis showed that the sample size of 128 allows for the detection of an effect size of 0.5 with a power of 0.8 when the two-sided Z-Test was used. The post hoc analysis based on the diagnosis of periodontal abscess was 0.9 with the current sample size (*N* = 122) when the two-sided Fisher’s exact test was used. 

### 2.4. Statistical Analysis

Once we obtained responses from the participants, the frequency data were generated as descriptive statistics. To examine the differences in the responses between the groups with respect to each item, a Fisher’s exact test was conducted. To explore and compare the flows of decision-making in diagnosis, prognosis, and treatment among the groups, alluvial diagrams were drawn using Jamovi (Version 1.6, Computer Software, Sydney, Australia) [11]. A *p*-value < 0.05 was considered significant.

## 3. Results

Table 3 shows the distributions of item responses with respect to each case among the participants. Only 31% of the participants diagnosed Case A as periodontal abscess. Fisher’s exact test for diagnosis and treatment items revealed that the distribution of item responses for Case A was significantly different between the senior 1 and DP groups. The most frequent diagnosis for Case A was periodontal abscess (44%) from the senior 1 group and root fracture (54%) from the DP group. While no GP diagnosed it as periapical abscess, 11% of the seniors diagnosed it as periapical abscess. The most frequent treatment option for Case A was periodontal surgery (65%) from the senior 1 group and extraction (63%) from the DP group. Regarding Case B, the most frequent diagnosis was periapical abscess from the senior 2 (75%) and the DP (73%) groups, showing the highest agreement among all responses to the items. However, the treatment choice was significantly different between the two groups: 57% of seniors selected root canal retreatment while 51% of DPs selected extraction for Case B. 

The three most frequent diagnoses for Case A were selected from the senior 1 and DP groups to draw alluvial plots. Figure 1 illustrates the distribution and correspondence of the diagnosis, prognosis, and treatment choice for Case A. The seniors, whose diagnosis was periodontal abscess, selected either surgical debridement or guided tissue regeneration (GTR) as the treatment option despite indicating a wide range of the prognosis assignment from good to questionable. Most treatment selections among the seniors, whose diagnosis was cyst, were still surgeries, such as enucleation, GTR, or surgical debridement with a similar range of the assigned prognosis. 80% of DPs, whose diagnosis was root fracture, assigned the hopeless prognosis and selected extraction. Extraction was still the treatment choice from half of DPs whose diagnosis was periodontal abscess although the fair or poor prognoses were assigned.

Figure 2 shows the distribution and correspondence of the diagnosis, prognosis, and treatment choice for Case B in the senior 2 and DP groups. Although > 70% participants diagnosed Case B as periapical abscess in the two groups, the distribution of the prognosis assignments was significantly different (Fisher’s exact test, *p* < 0.001). Eighty-five percent of seniors whose diagnosis was periapical abscess assigned the questionable prognosis and most of them selected root canal retreatment or root end surgery as the treatment option. Forty-five percent of GPs with the same diagnosis assigned either a hopeless or unfavorable prognosis and most of them selected extraction. While five DPs made a diagnosis of root fracture, no senior diagnosed it as root fracture.

Table 4 presents the distribution of the primary determinants for assigning a prognosis for each case from the senior 1, senior 2, and GP groups. While 92% of the senior 1 group responded with periodontal support as the primary determinant, the responses on the primary determinant were more diverse in the GP group for Case A. With respect to Case B, changes in the size of PARL was the most frequent primary determinant for assigning the prognosis of Case B among the participants.

## 4. Discussion 

Making a correct diagnosis for dental infection by identifying a source of the infection is important to preserve the prognosis of the affected tooth via delivering the proper treatment in a timely manner. However, discrepancies in diagnosis, interventional decisions, and treatment selections for individual teeth or the entire dentition among DPs have been reported [7,8]. This phenomenon was also observed in this study. 

While the confirmed diagnosis for Case A was periodontal abscess due to cemental tear, nine different diagnoses were made (Table 3). Only one DP out of all participants suspected cemental tear based on the clinical photographs obtained during the exploratory surgery. Although cone-beam computed tomography (CBCT) images along with the statement that root fracture did not appear to be evident were provided, 54% of DPs diagnosed Case A as root fracture. 

Cemental tear is a detachment of the cementum from the root surface, leading to rapid periodontal breakdown [12]. Root surface morphology, signs, and symptoms of cemental tear can be similar with those of root fracture [13]. Assuming their limited clinical experiences in cemental tear, most DPs might be confused with root fracture, which may be observed commonly in their practices. 

Seventy-four percent of participants made the correct working diagnosis for Case B (Table 3). Diagnosing a periapical abscess may be straightforward when a sinus tract is present and traced to the root apex in conjunction with the previous RCT compared to diagnosing periodontal abscesses with unknown pulpal diagnosis. However, 25% of seniors and 19% of DPs selected an endodontic-periodontal (endo-perio) combined lesion. This may indicate that a clinical reasoning error exists, especially in periodontal evaluation and interpretation. To diagnose an endo-perio combined lesion, intercommunication between pulpal and periodontal tissues via open structures should be confirmed [5]. Such intercommunication was hardly suspicious based on the maximum pocket depth of 4 mm and no furcation involvement in case B. 

Five DPs suspected a root fracture in Case B. Diagnosing root fracture is challenging because its clinical manifestations are inconsistent. Moreover, the visualization of the fracture line via either digital periapical radiography or CBCT is still limited with their low diagnostic sensitivity, 0.16 and 0.27, respectively [14,15]. Therefore, the diagnosis of root fracture should be considered with a combination of findings including radiographs, periodontal probings, sinus tracts and history of treatment. For example, the presence of the J-shaped radiographic halo around roots, which is a commonly seen in root fractures in many cases [16], is not a pathognomonic diagnostic finding for root fracture and is, at best, suggestive. Only 45% of vertical root fractures showed a J-shaped (apico-coronal-lateral) lesion; radiographic detection of vertical root fracture is possible only when the separation of root segments is evident or a “hair-like” fracture line can be detected [17].

This study examined the clinical reasoning process and flow with alluvial diagrams. Regarding Case A, most participants seemed to select an acceptable treatment for their corresponding diagnosis regardless of whether they made the correct diagnosis or not (Figure 1). The diagnosis also seemed to affect the selection of the primary determinant for assigning prognosis (Table 4). Forty-five percent of DPs assigned the hopeless prognosis largely based on non-periodontal parameters such as restorability or root fracture/morphology of the tooth. Ninety-two percent of seniors used periodontal support for assigning the prognosis, but, considering the high degree of its variability, the assigned prognosis might not influence the treatment selection in the senior 1 group. Regarding Case B, the combination of diagnosis and prognosis was deemed to affect the treatment selection because the choice of treatment was significantly different between the senior 2 and GP groups despite their high agreement in diagnosis. The distribution of assigned prognoses was significantly different between the two groups (Figure 2). 

Based on the alluvia diagrams, assigning a prognosis appears to be a more difficult task than making a diagnosis. Prognosis in healthcare fields commonly refers to the expected course of a disease or a condition and should be determined before formulating treatment plans [18]. The reason why DPs assigned a poorer prognosis than senior students and recommended more extraction warrants further study. 

Several periodontal prognostic models were developed and evaluated [9,10,19], but they might not depict the comprehensive individual tooth prognosis with their limited consideration for endodontic and restorative aspects [20]. While ample studies have reported the prognosis of initial endodontic therapy, the reported prognosis is diverse and difficult to interpret by clinicians due to their heterogeneities in outcome measurements, criteria, and classification along with a lack of pre-, intra-, and postoperative clinical and radiographic data [21,22]. Nevertheless, the presence of preoperative PARL was reported as a major indicator of endodontic postoperative healing or failure [23]. Clinicians still must be careful to determine the prognosis and choose appropriate treatment options with failed RCT. 

The generalizability of our findings is limited due to a few factors. First, this study could not evaluate the impact of the clinician’s knowledge/experience on decision-making in diagnosis, prognosis, and treatment selection. Measuring individual clinician’s knowledge level might not even be feasible. The clinicians’ experience may be reflected by the treating patient populations as well as years of practice and the postgraduate training history. Second, the study questionnaire contained closed-answer questions for the quantitative analysis, although the option of other was included. To investigate cognitive reasoning, open-ended or essay-type questions appear to be more appropriate [24]. Therefore, future studies should include clinicians’ experience levels and more essay-type questions for the qualitative analysis.

## 5. Conclusions

Only 31% of the participants correctly diagnosed the periodontal abscess, while 74% of the participants correctly diagnosed the periapical abscess. Based on the results of the study, the diagnosis of periodontal abscess was more challenging than that of periapical abscess among dental professionals.

## Figures and Tables

**Figure 1 healthcare-10-01393-f001:**
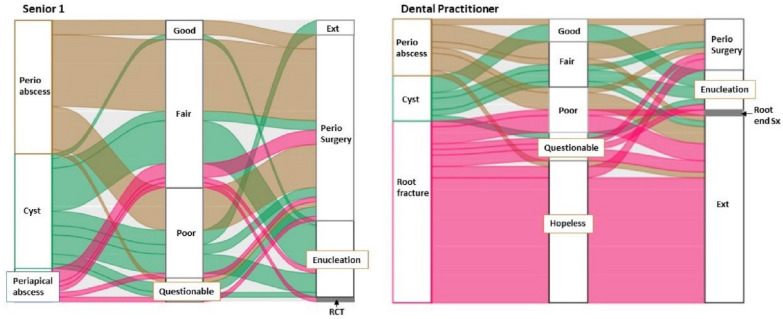
Distribution and correspondence of the diagnosis, prognosis, and treatment choice for Case A between the Senior 1 (*n* = 59) and Dental practitioner (*n* = 50) groups. Perio = periodontal, Ext = extraction, RCT = root canal therapy, Sx = surgery.

**Figure 2 healthcare-10-01393-f002:**
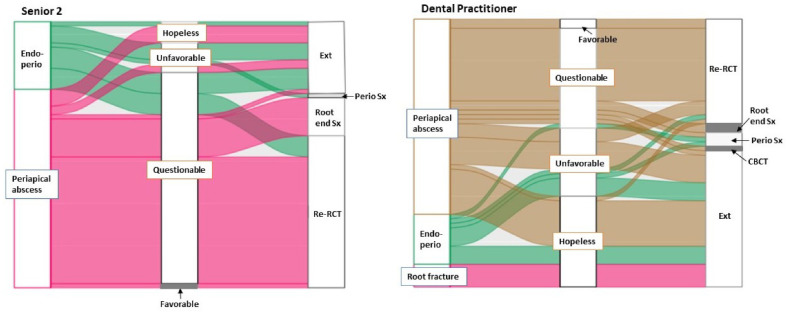
Distribution and correspondence of the diagnosis, prognosis, and treatment choice for Case B between the Senior 2 (*n* = 63) and Dental practitioner (*n* = 59) groups. Endo-perio = endodontic periodontal combined lesion, Ext = extraction, Sx = surgery, RCT = root canal therapy, CBCT = cone-beam computed tomography.

**Table 1 healthcare-10-01393-t001:** Questionnaire used in this study.

General Information	
Gender	○ Male
	○ Female
Postgraduate training	○ None
	○ Advanced general dentistry/General practice residency
	○ Periodontics
	○ Endodontics
	○ Others
**Case A**	
1. Based on the clinical and radiographical evaluation presented, what is the most appropriate diagnosis of the maxillary right central incisor?	
2. Based on the clinical and radiographical evaluation, assign the prognosis for the maxillary right central incisor using McGuire and Nunn prognosis system.	
3. What was your primary determinant for assigning the prognosis for the maxillary right central incisor?	
4. In your opinion, what is the most appropriate treatment option for this patient’s maxillary right central incisor?	
**Case B**	
1. Based on the clinical and radiographical evaluation presented, what is the most appropriate diagnosis of the mandibular right second molar?	
2. What is the most appropriate prognosis for the mandibular right second molar based on the clinical and radiographical evaluation? Use the Kwok and Caton prognosis scheme.	
3. What was your primary determinant for assigning the prognosis for the mandibular right second molar?	
4. In your opinion, what is the most appropriate treatment option for this patient’s mandibular right second molar?	

**Table 2 healthcare-10-01393-t002:** Summary of the participants (the number of participants).

Senior 1 (*n*1 = 63)	Senior 2 (*n*2 = 63)	Dental Practitioner (*n*3 = 59)
Male (29)	Male (29)	Male (39)
Female (34)	Female (34)	Female (20)
Final grade in the third-year periodontics (mean ± SD)		Postgraduate training
80.3 ± 12.2	79.2 ± 10	None (23) AGD/GPR (17) Endodontics (9) Periodontics (8) Others (2)

SD = standard deviation; AGD = advanced general dentistry; GPR = general practice residency.

**Table 3 healthcare-10-01393-t003:** Distribution of the responses to items from the senior 1, senior 2, and dental practitioner (DP) groups.

Case A	Senior 1 (*n*1 = 63)	DP (*n*3 = 59)	Fisher’s Exact Test
Diagnosis			
*Periodontal abscess* *	28 (44%)	10 (17%)	*p* < 0.001
Root fracture	1 (2%)	32 (54%)	
Cyst	24 (38%)	8 (14%)	
Other	10 (16%)	9 (15%)	
Periapical abscess	7	0	
Endodontic-periodontal combined lesion	1	3	
Invasion of biologic width	2	2	
Gingival abscess	0	1	
Necrotic pulp	0	2	
Periodontitis	0	1	
Treatment			
Root canal therapy	1 (2%)	1 (2%)	*p* < 0.001
Enucleation	17 (27%)	8 (13%)	
Extraction	4 (6%)	37 (63%)	
GTR/Surgical debridement	41 (65%)	7 (12%)	
**Case B**	**Senior 2** **(*n*2 = 63)**	**DP** **(*n*3 = 59)**	
Diagnosis			
*Periapical abscess* *	47 (75%)	43 (73%)	*p* = 0.05
Endodontic-periodontal combined lesion	16 (25%)	11 (19%)	
Root fracture	0 (0%)	5 (8%)	
Treatment			
Re-root canal therapy	36 (57%)	24 (41%)	*p* = 0.008
Extraction	17 (27%)	30 (51%)	
Periodontal Surgery	1 (2%)	3 (5%)	
Root-end Surgery	9 (14%)	2 (3%)	

* Correct working diagnosis; GTR = guided tissue regeneration.

**Table 4 healthcare-10-01393-t004:** Distribution of determinants for assigning a prognosis from the Senior 1, Senior 2, and Dental Practitioner (DP) groups. BL = bone loss, CAL = clinical attachment loss, PD = pocket depth, Dx = diagnosis, PARL = periapical radiolucency, RCT = root canal therapy.

Case A	Senior 1 (*n*1 = 63)	DP (*n*3 = 59)
Periodontal support (% BL/ CAL/ PD/ Periodontal Dx)	58 (92%)	27 (46%)
Restorability	2 (3%)	10 (17%)
Signs and symptom/ mobility	2 (3%)	8 (13.5%)
Root surface morphology	1 (2%)	7 (11.8%)
Root fracture		6 (10%)
Unknown pulpal Dx		1 (1.7%)
**Case B**	**Senior 2** **(*n*2 = 63)**	**DP** **(*n*3 = 59)**
Change in the size of PARL	29 (46%)	22 (37.3%)
Restorability	13 (20.6%)	4 (6.8%)
Periodontal support	11 (17.5%)	6 (10.2%)
Pulpal/periapical Dx	5 (7.9%)	17 (28.8%)
Success rate of treatment option	4 (6.3%)	6 (10.2%)
Failed RCT/ root fracture	1 (1.6%)	4 (6.8%)

## Data Availability

Not applicable.

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
