# Peer review of "Comparison Study of Diagnosis and Treatment Planning for Dental Infections between Dental Students and Practitioners"

_healthcare, 2022, doi:10.3390/healthcare10081393_

Round 1
Reviewer 1 Report
This is a very interesting manuscript and the topic is well covered. However, here are some comments required to revisit before considering further:
1. Were the Dental Practitioners classified according to degrees, such as DDS, DMD, or postgraduate degrees (MS, PhD etc). I was wondering whether there were any specialists or residents of periodontology, conservative dentistry, or endodontics as their knowledge might be slightly different?
2. Could you please clarify "seniors" among the students? As many countries have differences in curriculums and multiple years required for completion, such as first-year/final year or expected graduates?
Author Response
Dear Reviewer,
We appreciate your constructive feedback. We addressed your concerns as follows. Any changes in the main document were made with blue font.
- Were the Dental Practitioners classified according to degrees, such as DDS, DMD, or postgraduate degrees (MS, PhD etc). I was wondering whether there were any specialists or residents of periodontology, conservative dentistry, or endodontics as their knowledge might be slightly different?
The participating dental practitioners were all either DDS or DMD who are practicing at private dental clinics. Their postgraduate training information was listed in Table 2. We did not include postgraduate residents who are still under specialty training at school. Since the number of participating endodontists and periodontists was small, we did not compare their responses.
Table 2. Summary of the participants (the number of participants).
|
Senior 1 (n1 = 63) |
Senior 2 (n2 = 63) |
Dental Practitioner (n3 = 59) |
|
Male (29) |
Male (29) |
Male (39) |
|
Female (34) |
Female (34) |
Female (20) |
|
Final grade in the third-year periodontics (mean ± SD) |
Postgraduate training |
|
|
80.3 ± 12.2 |
79.2 ± 10 |
None (23) AGD/GPR (17) Endodontics (9) Periodontics (8) Others (2) |
SD = standard deviation; AGD = advanced general dentistry; GPR = general practice residency
- Could you please clarify "seniors" among the students? As many countries have differences in curriculums and multiple years required for completion, such as first-year/final year or expected graduates?
We added the term “final-year” to clarify “seniors”. Thank you for your comments.
Reviewer 2 Report
Dear authors,
1. This study aimed to access the knowledge in diagnosing dental infections and the practice in treatment planning for the affected teeth among dental practitioners and senior students.
2. The topic is interesting and addresses a gap related to the diagnosis of dental infections.
3. It presents the opinions of practitioners and students on 2 particular case reports
4. The methodology is based on a questionnaire. Practitioners and student opinions were comapred
5. Conclusions should be stated separately and clearly.
6. References should be improved. I suggest refining the references, there are ones dating from 1985, 1996, 2000, 2001, 2002, 2005.
7. I have recommended to improve Fig 1 and 2. Figures 1 and 2 should be made more understandable (the high coloristic makes them difficult to understand).
Author Response
Dear Reviewer,
We appreciate your constructive feedback. We addressed your concerns as follows. Any changes in the main text were made with blue font.
1. This study aimed to access the knowledge in diagnosing dental infections and the practice in treatment planning for the affected teeth among dental practitioners and senior students.
2. The topic is interesting and addresses a gap related to the diagnosis of dental infections.
3. It presents the opinions of practitioners and students on 2 particular case reports
4. The methodology is based on a questionnaire. Practitioners and student opinions were compared.
Thank you for all your comments from 1 to 4.
5. Conclusions should be stated separately and clearly.
Conclusions were stated as follows.
CONCLUSION
Only 31% of the participants correctly diagnosed the periodontal abscess while 74% of the participants correctly diagnosed the periapical abscess. Based on the results of the study, diagnosis of periodontal abscess was more challenging than that of periapical abscess among dental professionals.
6. References should be improved. I suggest refining the references, there are ones dating from 1985, 1996, 2000, 2001, 2002, 2005.
We refined the references from 1985, 2001, 2002, and 2005. We did not update the following references because the McGuire and Nunn study presenting the prognosis system cannot be replaced by another one and Herrara et al article is the one that included all of the prevalence that was listed in this manuscript.
McGuire, M. K.; Nunn, M. E. Prognosis Versus Actual Outcome. II. The Effectiveness of Clinical Parameters in Developing an Accurate Prognosis. J. Periodontol. 1996, 67, 658–665.
Herrera, D.; Roldán, S.; Sanz, M. The Periodontal Abscess: A Review. J. Clin. Periodontol. 2000, 27, 377–386.
7. I have recommended to improve Fig 1 and 2. Figures 1 and 2 should be made more understandable (the high coloristic makes them difficult to understand).
Both figures were revised to reduce colors.
Thank you.